# DeepV2D: Video to Depth with Differentiable Structure from Motion

**Zachary Teed**
Princeton University
zteed@cs.princeton.edu

**Jia Deng**
Princeton University
jiadeng@cs.princeton.edu

## Abstract

We propose DeepV2D, an end-to-end deep learning architecture for predicting depth from video. DeepV2D combines the representation ability of neural networks with the geometric principles governing image formation. We compose a collection of classical geometric algorithms, which are converted into trainable modules and combined into an end-to-end differentiable architecture. DeepV2D interleaves two stages: motion estimation and depth estimation. During inference, motion and depth estimation are alternated and converge to accurate depth. Code is available https://github.com/princeton-vl/DeepV2D.

## 1 Introduction

In video to depth, the task is to estimate depth from a video sequence. The problem has traditionally been approached using Structure from Motion (SfM), which takes a collection of images as input, and jointly optimizes over 3D structure and camera motion (Schonberger & Frahm, 2016b). The resulting camera parameter estimates can be used as input to Multi-View Stereo in order to build a more complete 3D representation such as surface meshes and depth maps (Furukawa et al., 2015; Furukawa & Ponce, 2010).

In parallel, deep learning has been highly successful in a number of 3D reconstruction tasks. In particular, given ground truth depth, a network can learn to predict depth from a single image (Eigen et al., 2014; Eigen & Fergus, 2015; Laina et al., 2016), stereo images (Kendall et al., 2017; Mayer et al., 2016a), or collections of frames (Zhou et al., 2018; Kar et al., 2017; Tang & Tan, 2018; Yao et al., 2018). One advantage of deep networks is that they can use single-image cues such as texture gradients and shading as shown by their strong performance on depth estimation from a single image (Eigen et al., 2014; Eigen & Fergus, 2015; Laina et al., 2016). Furthermore, differentiable network modules can be composed so that entire pipelines (i.e. feature extraction, feature matching, regularization) can be learned directly from training data. On the other hand, as recent work has shown, it is often hard to train generic network layers to directly utilize multiview geometry (e.g. using interframe correspondence to recover depth), and it is often advantageous to embed knowledge of multiview geometry through specially designed layers or losses (Ummenhofer et al., 2017; Kendall & Cipolla, 2017; Zhou et al., 2017; Vijayanarasimhan et al., 2017; Zhou et al., 2018).

In this work, we continue the direction set forth by recent works (Ummenhofer et al., 2017; Kendall et al., 2017; Tang & Tan, 2018; Zhou et al., 2018; Kar et al., 2017; Wang et al., 2018) that combine the representation ability of neural networks with the geometric principles underlying image forma-

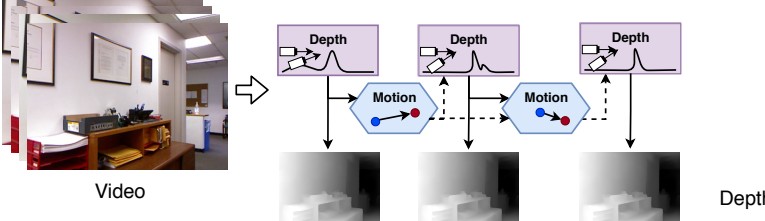

Figure 1: DeepV2D predicts depth from video. It is the composition of classical geometric algorithms, made differentiable, and combined into an end-to-end trainable network architecture. Video to depth is broken down into the subproblems of motion estimation and depth estimation, which are solved by the Motion Module and Depth Module respectively.

tion. We propose DeepV2D, a composition of classical geometrical algorithms which we turn into differentiable network modules and combine into an end-to-end trainable architecture. DeepV2D interleaves two stages: camera motion estimation and depth estimation (Figure 1). The motion module takes depth as input, and outputs an incremental update to camera motion. The depth module takes camera motion as input, and performs stereo reconstruction to predict depth. At test time, DeepV2D acts as block coordinate descent, alternating between updating depth and camera motion.

To estimate camera motion we introduce Flow-SE3, a new motion estimation architecture, which outputs an incremental update to camera motion. Flow-SE3 takes depth as input, and estimates dense 2D correspondence between pairs of frames. We unroll a single iteration of Perspective-n-Point (PnP) (Lepetit et al., 2009; Li et al., 2012) performing Gauss-Newton updates over SE3 perturbations to minimize geometric reprojection error. The new estimate of camera motion can then be fed back into Flow-SE3, which re-estimates correspondence for a finer grain pose update.

Our Depth Module builds upon prior work (Kendall et al., 2017; Yao et al., 2018) and formulates multiview-stereo (MVS) reconstruction as a single feed-forward network. Like classical MVS, we leverage geometry to build a cost volume over video frames, but use trainable network for both feature extraction and matching.

Our work shares similarities with prior works (Ummenhofer et al., 2017; Kendall et al., 2017; Tang & Tan, 2018; Zhou et al., 2018; Kar et al., 2017; Wang et al., 2018) that also combine deep learning and multiview geometry, but is novel and unique in that it essentially "differentializes" a classical SfM pipeline that alternates between stereopsis, dense 2D feature matching, and PnP. As a comparison, DeMon (Ummenhofer et al., 2017) and DeepTAM (Zhou et al., 2018) differentialize stereopsis and feature matching, but not PnP because they use a generic network to predict camera motion.

Another comparison is with BA-Net (Tang & Tan, 2018), whose classical analogue is performing bundle adjustment from scratch to optimize feature alignment over camera motion and the coefficients of a limited set of depth maps (depth basis). In other words, BA-Net performs one joint nonlinear optimization over all variables, whereas we decompose the joint optimization into more tractable subproblems and do block coordinate descent. Our decomposition is more expressive in terms of reconstruction since we can optimize directly over per-pixel depth and are not constrained by a depth basis, which can potentially limit the accuracy of the final depth.

In our experiments, we demonstrate the effectiveness of DeepV2D across a variety of datasets and tasks, and outperform strong methods such as DeepTAM (Zhou et al., 2018), DeMoN (Ummenhofer et al., 2017), BANet (Tang & Tan, 2018), and MVSNet (Yao et al., 2018). As we show, alternating depth and motion estimation quickly converges to good solutions. On all datasets we outperform all existing single-view and multi-view approaches. We also show superior cross-dataset generalizability, and can outperform existing methods even when training on entirely different datasets.

## 2 RELATED WORK

**Structure from Motion:** Beginning with early systems designed for small image collections (Longuet-Higgins, 1981; Mohr et al., 1995), Structure from Motion (SfM) has improved dramatically in regards to robustness, accuracy, and scalability. Advances have come from improved features (Lowe, 2004; Han et al., 2015), optimization techniques (Snavely, 2009), and more scalable data structures and representations (Schonberger & Frahm, 2016a; Gherardi et al., 2010), culminating in a number of robust systems capable of large-scale reconstruction task (Schonberger & Frahm, 2016a; Snavely, 2011; Wu et al., 2011). Ranftl et al. (2016) showed that SfM could be extended to reconstruct scenes containing many dynamically moving objects. However, SfM is limited by the accuracy and availability of correspondence. In low texture regions, occlusions, or lighting changes SfM can produce noisy or missing reconstructions.

Simultaneous Localization and Mapping (SLAM) jointly estimates camera motion and 3D structure from a video sequence (Engel et al., 2014; Mur-Artal et al., 2015; Mur-Artal & Tardós, 2017; Newcombe et al., 2011; Engel et al., 2018). LSD-SLAM (Engel et al., 2014) is unique in that it relies on a featureless approach to 3D reconstruction, directly estimating depth maps and camera pose by minimizing photometric error. Our Motion Network behaves similarly to the tracking component in LSD-SLAM, but we use a network which predicts misalignment directly instead of using intensity gradients. We end up with an easier optimization problem characteristic of indirect methods (Mur-

Artal et al., 2015), while retaining the flexibility of direct methods in modeling edges and smooth intensity changes (Engel et al., 2018).

**Geometry and Deep Learning:** Geometric principles has motivated the design of many deep learning architectures. In video to depth, we need to solve two subproblems: depth estimation and motion estimation.

*Depth*: End-to-end networks can be trained to predict accurate depth from a rectified pair of stereo images (Han et al., 2015; Mayer et al., 2016a; Kendall et al., 2017; Chang & Chen, 2018). Kendall et al. (2017) and Chang & Chen (2018) design network architectures specifically for stereo matching. First, they apply a 2D convolutional network to extract learned features, then build a cost volume over the learned features. They then apply 3-D convolutions to the cost volume to perform feature matching and regularization. A similar idea has been extended to estimate 3D structure from multiple views (Kar et al., 2017; Yao et al., 2018). In particular, MVSNet (Yao et al., 2018) estimates depth from multiple images. However, these works require known camera poses as input, while our method estimates depth from a video where the motion of the camera is unknown and estimated during inference.

*Motion*: Several works have used deep networks to predict camera pose. Kendall et al. (2015) focus on the problem of camera localization, while other work (Zhou et al., 2017; Vijayanarasimhan et al., 2017; Wang et al., 2017) propose methods which estimate camera motion between a pairs of frames in a video. Networks for motion estimation have typically relied on generic network components whereas we formulate motion estimation as a least-squares optimization problem. Whereas prior work has focused on estimating relative motion between pairs of frames, we can jointly update the pose of a variable number of frames.

*Depth and Motion*: Geometric information has served as a self-supervisory signal for many recent works (Vijayanarasimhan et al., 2017; Zhou et al., 2017; Wang et al., 2018; Yin & Shi, 2018; Yang et al., 2018; Godard et al., 2017; Mahjourian et al., 2018). In particular, Zhou et al. (2017) and Vijayanarasimhan et al. (2017) trained a single-image depth network and a pose network while supervising on photometric consistency. However, while these works use geometric principles for training, they do not use multiple frames to predict depth at inference.

DeMoN (Ummenhofer et al., 2017) and DeepTAM (Zhou et al., 2018) where among the first works to combine motion estimation and multi-view reconstruction into a trainable pipeline. DeMoN (Ummenhofer et al., 2017) operates on two frames and estimates depth and motion in separate network branches, while DeepTAM (Zhou et al., 2018) can be used on variable number of frames. Like our work and other classical SLAM frameworks (Engel et al., 2014; Newcombe et al., 2011), Deep-TAM separates depth and motion estimation, however we maintain end-to-end differentiablity between our modules. A major innovation of DeepTAM was to formulate camera motion estimation in the form of incremental updates. In each iteration, DeepTAM renders the keyframe from a synthetic viewpoint, and predicts the residual motion from the rendered viewpoint and the target frame.

Estimating depth and camera motion can be naturally modeled as a non-linear least squares problem, which has motivated several works to include an differentiable optimization layer within network architectures (Tang & Tan, 2018; Wang et al., 2018; Clark et al., 2018; Bloesch et al., 2018). We follow this line of work, and propose the Flow-SE3 module which introduces a direct mapping from 2D correspondence to a 6-dof camera motion update. Our Flow-SE3 module is different from prior works such as DeMon (Ummenhofer et al., 2017) and DeepTAM (Zhou et al., 2018) which do not impose geometric constraints on camera motion and use generic layers. BA-Net (Tang & Tan, 2018) and LS-Net (Clark et al., 2018) include optimization layers, but instead optimize over photometric error (either pixel alignment (Clark et al., 2018) or feature alignment (Tang & Tan, 2018)). Our Flow-SE3 module still imposes geometric constraints on camera motion like BA-Net (Tang & Tan, 2018), but we show that in minimizing geometric reprojection error ( difference of pixel locations), we end up with a well-behaved optimization problem, well-suited for end-to-end training.

An important difference between our approach and BA-Net is that BA-Net performs one joint optimization problem by formulating Bundle-Adjustment as a differentiable network layer, whereas we separate motion and depth estimation. With this separation, we avoid the need for a depth basis. Our final reconstructed depth is the product of a cost volume, which can adapt the reconstruction as camera motion updates improve, while the output of BA-Net is restricted by the initial quality of the depth basis produced by a single-image network.

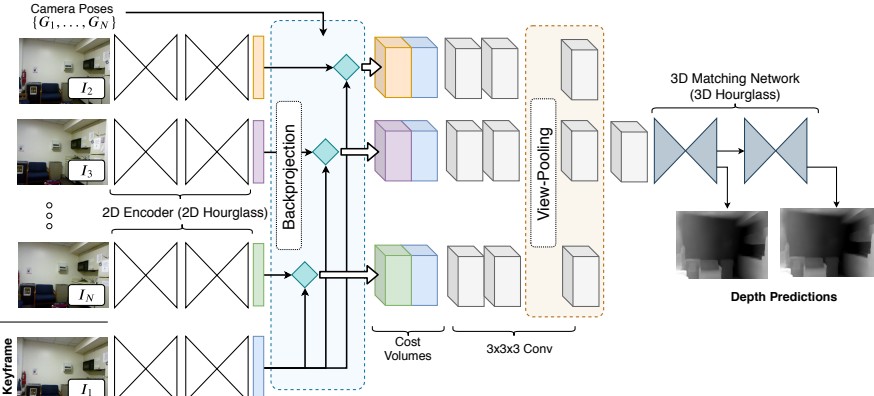

Figure 2: The *Depth Module* performs stereo matching over multiple frames to estimate depth. First each image is fed through a network to extract a dense feature map. The 2D features are backprojected into a set of cost volumes. The cost volumes are processed by a set of 3D hourglass networks to perform feature matching. The final cost volume is processed by the differentiable arg-max operator to produce a pixelwise depth estimate.

## 3 APPROACH

DeepV2D predicts depth from a calibrated video sequence. We take a video as input and output dense depth. We consider two subproblems: depth estimation and motion estimation. Both subproblems are formulated as trainable neural network modules, which we refer to as the *Depth Module* and the *Motion Module*. Our depth module takes camera motion as input and outputs an updated depth prediction. Our motion module takes depth as input, and outputs a motion correction term. In the forward pass, we alternate between the depth and motion modules as we show in Figure 1.

**Notation and Camera Geometry:** As a preliminary, we define some of the operations used within the depth and motion modules. We define $\pi$ to be the camera projection operator which maps a 3D point $\mathbf{X} = (X, Y, Z, 1)^T$ to image coordinates $\mathbf{x} = (u, v)$. Likewise, $\pi^{-1}$ is defined to be the backprojection operator, which maps a pixel $x$ and depth $z$ to a 3D point. Using the pinhole camera model with intrinsics $(f_x, f_y, c_x, c_y)$ we have

$$\pi(\mathbf{X}) = (f_x \frac{X}{Z} + c_x, f_y \frac{Y}{Z} + c_y), \qquad \pi^{-1}(\mathbf{x}, z) = (z \frac{u - c_x}{f_x}, z \frac{v - c_y}{f_y}, z, 1)^T \qquad (1)$$

The camera pose is represented using rigid body transform $\mathbf{G} \in SE(3)$. To find the image coordinates of point $\mathbf{X}$ in camera $i$, we chain the projection and transformation: $(u, v)^T = \pi(\mathbf{G}_i \mathbf{X})$, where $\mathbf{G}_i$ is the pose of camera $i$.

Now, given two cameras $\mathbf{G}_i$ and $\mathbf{G}_j$. If we know the depth of a point $\mathbf{x}^i = (u^i, v^i)$ in camera $i$, we can find its reprojected coordinates in camera $j$:

$$\begin{pmatrix} u^j \\ v^j \end{pmatrix} = \pi(\mathbf{G}_j \mathbf{G}_i^{-1} \pi^{-1}(\mathbf{x}, z)) = \pi(\mathbf{G}_{ij} \pi^{-1}(\mathbf{x}, z)) \qquad (2)$$

using the notation $\mathbf{G}_{ij} = \mathbf{G}_j \mathbf{G}_i^{-1}$ for the relative pose between cameras $i$ and $j$.

### 3.1 DEPTH MODULE

The depth module takes a collection of frames, $\mathbf{I} = \{I_1, I_2, ..., I_N\}$, along with their respective pose estimates, $\mathbf{G} = \{G_1, G_2, ..., G_N\}$, and predicts a dense depth map $D^*$ for the keyframe (Figure 2). The depth module works by building a cost volume over learned features. Information is aggregated over multiple viewpoints by applying a global pooling layer which pools across viewpoints.

The depth module can be viewed as the composition of 3 building blocks: 2D feature extractor, cost volume backprojection, and 3D stereo matching.

**2D Feature Extraction:** The Depth Module begins by extracting learned features from the input images. The 2D encoder consists of 2 stacked hourglass modules (Newell et al., 2016) which maps

each image to a dense feature map $I_i \rightarrow F_i$. More information regarding network architectures is provided in the appendix.

**Cost Volume Backprojection:** Take $I_1$ to be the keyframe, a cost volume is constructed for each of the remaining N-1 frames. The cost volume for frame $j$, $\mathbf{C}^j$, is constructed by backprojecting 2D features into the coordinate system defined by the keyframe image. To build the cost volume, we enumerate over a range of depths $z_1, z_2, ..., z_D$ which is chosen to span the ranges observed in the dataset (0.2m - 10m for indoor scenes). For every depth $z_k$, we use Equation 2 to find the reprojected coordinates on frame $j$, and then use differentiable bilinear sampling of the feature map $F_j$.

More formally, given a pixel $\mathbf{x} = (u, v) \in \mathbb{N}^2$ in frame $I_1$ and depth $z_k$:

$$C_{uvk}^j = F_j(\pi(\mathbf{G}_j \mathbf{G}_1^{-1} \pi^{-1}(\mathbf{x}, z_k))) \in \mathbb{R}^{H \times W \times D \times C} \qquad (3)$$

where $F(\cdot)$ is the differentiable bilinear sampling operator (Jaderberg et al., 2015). Since the bilinear sampling is differentiable, $\mathbf{C}^j$ is differentiable w.r.t all inputs, including the camera pose.

Applying this operation to each frame, gives us a set of N-1 cost volumes each with dimension H×W×D×C. As a final step, we concatenate each cost volume with the keyframe image features increasing the dimension to H×W×D×2C. By concatenating features, we give the network the necessary information to perform feature matching between the keyframe/image pairs without decimating the feature dimension.

**3D Matching Network:** The set of N-1 cost volumes are first processed by a series of 3D convolutional layers to perform stereo matching. We then perform view pooling by averaging over the N-1 volumes to aggregate information across frames. View pooling leaves us with a single volume of dimension H×W×D×C. The aggregated volume is then processed by a series of 3D hourglass modules, each outputs an intermediate depth.

Each 3D hourglass module predicts an intermediate depth estimate. We produce an intermediate depth representation by first applying a 1x1x1 convolution to a produce H×W×D volume. We then apply the softmax operator over the depth dimension, so that for each pixel, we get a probability distribution over depths. We map the probability volume into a single depth estimate using the differentiable argmax function (Kendall et al., 2017) which computes the expected depth.

## 3.2 MOTION MODULE

The objective of the motion module is to update the camera motion estimates given depth as input. Given the input poses, $\mathbf{G} = \{G_1, G_2, ..., G_N\}$, the motion module outputs a set of local perturbations $\boldsymbol{\xi} = \{\xi_1, \xi_2, ..., \xi_N\}, \xi_i \in se(3)$ used to update the poses. The updates are found by setting up a least squares optimization problem which is solved using a differentiable in-network optimization layer.

**Initialization:** We use a generic network architecture to predict the initial pose estimates similiar to prior work Zhou et al. (2017). We choose one frame to be the keyframe. The poses are initialized by setting the keyframe pose to be the identity matrix, and then predicting the relative motion between the keyframe and each of the other frames in the video.

**Feature Extraction:** Our motion module operates over learned features. The feature extractor maps every frame to a dense feature map, $I_i \rightarrow F_i$. The weights of the feature extractor are shared across all frames. Network architecture details are provided in the appendix.

**Error Term:** Take two frames, $(I_i, I_j)$, with respective poses $(\mathbf{G}_i, \mathbf{G}_j)$ and feature maps $(F_i, F_j)$. Given depth $Z_i$ we can use Equation 2 we can warp $F_j$ onto camera $i$ to generate the warped feature map $\tilde{F}_j$. If the relative pose $\mathbf{G}_{ij} = \mathbf{G}_j \mathbf{G}_i^{-1}$ is correct, then the feature maps $F_i$ and $\tilde{F}_j$ should align. However, if the relative pose is noisy, then there will be misalignment between the feature images which should be corrected by the pose update.

We concatenate $F_i$ and $\tilde{F}_j$, and send the concatenated feature map through an hourglass network to predict the dense residual flow between the feature maps, which we denote $\mathbf{R}$, and corresponding confidence map $\mathbf{W}$. Using the residual flow, we define the following error term:

$$\mathbf{e}_k^{ij}(\xi_i, \xi_j) = \mathbf{r}_k - [\pi((e^{\xi_j} \mathbf{G}_j)(e^{\xi_i} \mathbf{G}_i)^{-1} \mathbf{X}_k^i) - \pi(\mathbf{G}_{ij} \mathbf{X}_k^i)], \qquad \mathbf{X}_k^i = \pi^{-1}(\mathbf{x}_k, z_k) \qquad (4)$$

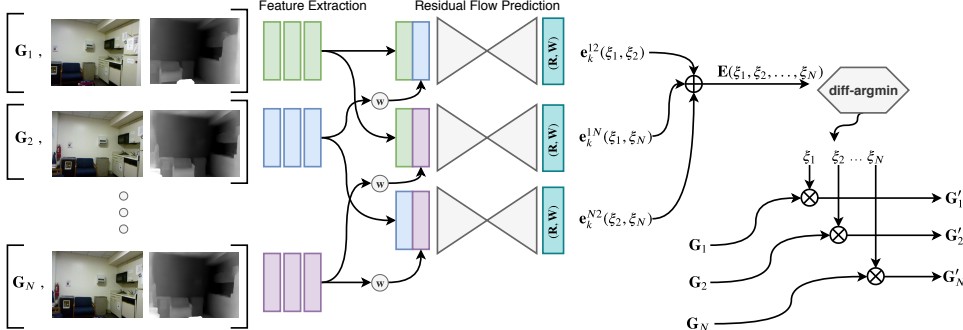

Figure 3: The *Motion Module* updates the input pose estimates by solving a least squares optimization problem. The motion module predicts the residual flow between pairs of frames, and uses the residual terms to define the optimization objective. Pose increments $\boldsymbol{\xi}$ are found by performing a single differentiable Gauss-Newton optimization step.

where $\mathbf{r}_k$ is the residual flow at pixel $\mathbf{x}_k$ predicted by the network, and $z_k$ is the predicted depth. The weighting map $\mathbf{W}$ is mapped to $(0, 1)$ using the sigmoid activation, and is used to determine how the individual error terms are weighted in the final objective.

**Optimization Objective:** The previous section showed how two frames $(i, j)$ can be used to define a collection of error terms $\mathbf{e}_k^{ij}(\xi_i, \xi_j)$ for each pixel $\mathbf{x}_k$ in image $I_i$. The final optimization objective is a weighted combination of error terms:

$$E(\boldsymbol{\xi}) = \sum_{(i,j)\in\mathcal{C}} \sum_k \mathbf{e}_k^{ij}(\xi_i, \xi_j)^T \, diag(\mathbf{w}_k) \, \mathbf{e}_k^{ij}(\xi_i, \xi_j), \qquad diag(\mathbf{w}_k) = \begin{pmatrix} w_k^u & 0 \\ 0 & w_k^v \end{pmatrix} \qquad (5)$$

This leaves us with the question of which frames pairs $(i, j) \in \mathcal{C}$ to use when defining the optimization objective. In this paper, we consider two different approaches which we refer to as *Global* pose optimization and *Keyframe* pose optimization.

***Global Pose Optimization:*** Our global pose optimization uses all pairs of frames $\mathcal{C} = (i, j), i \neq j$ to define the objective function (Equation 5) and the pose increment $\xi$ is solved for jointly over all poses. Therefore, given $N$ frames, dense pose optimization uses N×N-1 frame pairs. Since every pair of frames is compared, this means that the global pose optimization requires the predicted depth maps for all frames as input. Although each pair $(i, j)$ only gives us information about the relative pose $\mathbf{G}_{ij}$, considering all pairs allows us to converge to a globally consistent pose graph.

***Keyframe Pose Optimization:*** Our keyframe pose optimization selects a given frame to be the keyframe (i.e select $I_1$ as the keyframe), and only computes the error terms between the keyframe and each of the other frames: $\mathcal{C} = (1, j)$ for $j = 2, ..., N$.

Fixing the pose of the keyframe, we can remove $\xi_1$ from the optimization objective. This means that each error $\mathbf{e}_k^{ij}(\mathbf{0}, \xi_j)$ term is only a function of a single pose increment $\xi_j$. Therefore, we can solve for each of the $N - 1$ pose increments independently. Additionally, since $i = 1$ for all pairs $(i, j) \in \mathcal{C}$, we only need the depth of the keyframe as input when using *keyframe* pose optimization.

**LS-Optimization Layer:** Using the optimization objective in Equation 5, we solve for the pose increments $\boldsymbol{\xi}$ by applying a Gauss-Newton update. We backpropagate through the Gauss-Newton update so that the weights of the motion module (both feature extractor and flow network) can be trained on the final objective function. In the appendix, we provide additional information for how the update is derived and the expression for the Jacobian of Equation 4.

### 3.3 FULL SYSTEM

During inference, we alternate the depth and motion modules for a selected number of iterations. The motion module uses depth to predict camera pose. As the depth estimates converge, the camera poses become more accurate. Likewise, as camera poses converge, the depth module can estimate more accurate depth.

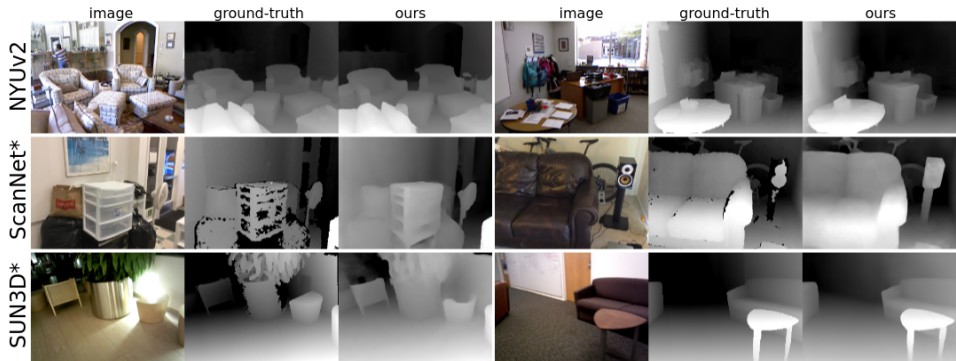

Figure 4: Visualization of predicted depth maps on NYU, ScanNet, and SUN3D. On ScanNet and SUN3D (marked with *) we show the results of the model trained only on NYU data.

**Initialization:** We try two different strategies for initialization in our experiments: (1) self initialization initializes DeepV2D with a constant depth map and (2) single image initialization uses the output of a single-image depth network for initialization. Both methods give good performance.

### 3.4 SUPERVISION

**Depth Supervision:** We supervise on the L1 distance between the ground truth and predicted depth. We additionally apply a small L1 smoothness penalty to the predicted depth map. Given predicted depth $Z$ and ground truth depth $Z^*$, the depth loss is defined as:

$$\mathcal{L}_{depth}(Z) = \sum_{\mathbf{x}_i} |Z(\mathbf{x}_i) - Z^*(\mathbf{x_i})| + w_s \sum_{\mathbf{x}_i} |\partial_x Z(\mathbf{x}_i)| + |\partial_y Z(\mathbf{x}_i)| \tag{6}$$

**Motion Supervision:** We supervise pose using the geometric reprojection error. Given predicted pose $\mathbf{G}$ and ground truth pose $\mathbf{G}^*$, the pose loss is defined

$$\mathcal{L}_{motion}(\mathbf{G}) = \sum_{\mathbf{x}_i} ||\pi(\mathbf{G}\pi^{-1}(\mathbf{x}_i, Z(\mathbf{x}_i))) - \pi(\mathbf{G}^*\pi^{-1}(\mathbf{x}_i, Z(\mathbf{x}_i)))||_\delta \tag{7}$$

where $|| \cdot ||_\delta$ is the robust Huber loss; we set $\delta = 1$.

**Total Loss:** The total loss is taken as a weighted combination of the depth and motion loss terms: $\mathcal{L} = \mathcal{L}_{depth} + \lambda \mathcal{L}_{motion}$, where we set $\lambda = 1.0$ in our experiments.

## 4 EXPERIMENTS

We test DeepV2D across a wide range of benchmarks to provide a thorough comparison to other methods. While the primary focus of these experiments is to compare to other works which estimate depth from multiple frames, often single-view networks still outperform multiview depth estimation. To put our results in proper context, we include both multiview and state-of-the-art single-image comparisons. Since it is not possible to recover the absolute scale of the scene through SfM, we report all results (both ours and all other approaches) using scale matched depth (Tang & Tan, 2018).

Our primary experiments are on NYU, ScanNet, SUN3D, and KITTI, and we report strong results across all datasets. We show visualization of our predicted depth maps in Figure 4. The figure shows that DeepV2D can recover accurate and sharp depth even when applied to unseen datasets. One aspect of particular interest is cross-dataset generalizability. Our results show that DeepV2D generalizes very well—we achieve the highest accuracy on ScanNet and SUN3D even without training on either dataset.

### 4.1 DEPTH EXPERIMENTS

We evaluate depth on NYU (Silberman et al., 2012), ScanNet (Dai et al., 2017), SUN3D (Xiao et al., 2013), and KITTI (Geiger et al., 2013). On all datasets, DeepV2D is given a video clip with unknown camera poses and alternates depth and pose updates and is evaluated after 8 iterations.

**NYU:** NYU depth (Silberman et al., 2012) is a dataset composed of videos taken in indoor settings including offices, bedrooms, and libraries. We experiment on NYU using the standard train/test split (Eigen et al., 2014) and report results in Table 1 using scaled depth (Zhou et al., 2017; Tang & Tan, 2018). We evaluate two different initialization methods of our approach. Self-init uses a constant depth map for initialization, while fcrn-init uses the output of a FCRN (Laina et al., 2016)—a single-view network for initialization. Using a single-image depth network for initialization gives a slight improvement in performance.

| | NYUv2 | $\delta < 1.25 \uparrow$ | $\delta < 1.25^2 \uparrow$ | $\delta < 1.25^3 \uparrow$ | Abs Rel $\downarrow$ | Sc Inv $\downarrow$ | RMSE $\downarrow$ | log10 $\downarrow$ |
|---|---|---|---|---|---|---|---|---|
| single | FCRN (Laina et al., 2016) | 0.853 | 0.965 | 0.991 | 0.121 | 0.151 | 0.592 | 0.052 |
| | DORN (Fu et al., 2018) | 0.875 | 0.966 | 0.989 | 0.109 | - | 0.464 | 0.047 |
| | Alhashim & Wonka (2018) | 0.895 | 0.980 | 0.996 | 0.103 | - | **0.390** | 0.043 |
| multi-view | COLMAP | 0.619 | 0.760 | 0.829 | 0.312 | 1.512 | 1.381 | 0.153 |
| | DfUSMC | 0.487 | 0.697 | 0.814 | 0.447 | 0.456 | 1.793 | 0.169 |
| | MVSNet + OpenMVG | 0.766 | 0.913 | 0.965 | 0.181 | 0.212 | 0.917 | 0.072 |
| | DeMoN | 0.776 | 0.933 | 0.979 | 0.160 | 0.196 | 0.775 | 0.067 |
| | DeMoN † | 0.805 | 0.951 | 0.985 | 0.144 | 0.179 | 0.717 | 0.061 |
| | Ours (self-init) - Keyframe | 0.940 | 0.985 | 0.995 | 0.072 | 0.105 | 0.459 | 0.031 |
| | Ours (fcrn-init) - Keyframe | 0.955 | **0.990** | **0.996** | 0.062 | 0.095 | 0.405 | 0.027 |
| | Ours (self-init) - Global | 0.942 | 0.986 | 0.995 | 0.070 | 0.104 | 0.454 | 0.030 |
| | Ours (fcrn-init) - Global | **0.956** | 0.989 | **0.996** | **0.061** | **0.094** | 0.403 | **0.026** |

Table 1: Results on the NYU dataset. Our approach outperforms existing single-view and multi-view depth estimation methods. Ours (self-init) uses a constant depth map for initialization while ours(fcrn-init) uses a single-image depth network for initialization.

We compare to state-of-the-art single-image depth networks DORN (Fu et al., 2018) and DenseDepth (Alhashim & Wonka, 2018) which are built on top of a pretrained ResNet (DORN) or DenseNet-201 (DenseDepth). The results show that we can do much better than single-view depth by using multiple views. We also include classical multiview approaches such as COLMAP (Schonberger & Frahm, 2016a) and DfUSMC (Ha et al., 2016) which estimate poses with bundle adjustment, followed by dense stereo matching. While COLMAP uses SIFT features, DfUSMC is built on local-feature tracking and is designed for small baseline videos.

Table 1 also includes results using multi-view deep learning approaches. MVSNet (Yao et al., 2018) is trained to estimate depth from multiple viewpoints. Unlike our approach which estimates camera pose during inference, MVSNet requires ground truth poses as input. We train MVSNet on NYU and use poses estimated from OpenMVG (Moulon et al.) during inference. Finally, we also evaluate DeMoN (Ummenhofer et al., 2017) on NYU. DeMoN is not originally trained on NYU, but instead trained on a combination of 5 other datasets. We also try a version of DeMoN which we retrain on NYU using the code provided by the authors (denoted †).

In Appendix C, we include additional results on NYU where we test different versions of our model, along with parameter counts, timing information, peak memory usage, and depth accuracy. A shallower version of DeepV2D (replacing the stacked hourglass networks with a single hourglass network) and lower resolution inference still outperform existing work on NYU. However, using a 3D network for stereo matching turns out to be very important for depth accuracy. When the 3D stereo network is replaced with a correlation layer (Dosovitskiy et al., 2015) and 2d encoder-decoder, depth accuracy is worse increasing Abs-Rel from 0.062 to 0.135.

Figure 5 shows the impact of the number of iterations and views on the scale-invariant (sc-inv) validation set accuracy. Figure 5 (left) shows that DeepV2D requires very few iterations to converge, suggesting that block coordinate descent is effective for estimate depth from small video clips. In Figure 5 (right) we test accuracy as a function of the number of input frames used. Although DeepV2D is trained using a fixed number (4) frames as input, accuracy continues to improve a more frames are added.

**ScanNet:** ScanNet is a large indoor dataset consisting of 1513 RGB-D videos in distinct scenes. We use the train/test split proposed by Tang & Tan (2018) and evaluate depth and pose accuracy in Table 2. While our primary focus is on depth, DeepV2D accurately predicts camera motion.

We use ScanNet to test cross-dataset generalization and report results from two versions of our approach: ours (nyu) is our method trained only on nyu, ours (scannet) is our method trained on ScanNet. As expected, when we train on the ScanNet training set we do better than if we train only on NYU. But the performance of our NYU model is still good and outperforms BA-Net on

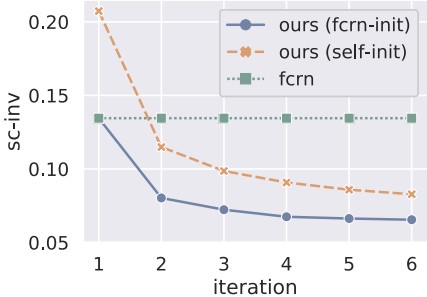 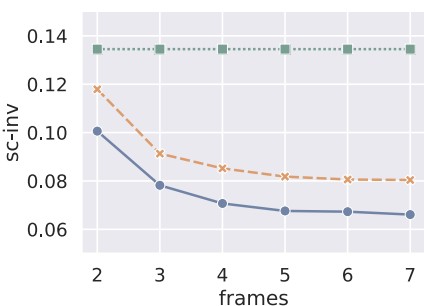

Figure 5: Impact of the number of iterations (left) and frames (right) on sc-inv validation accuracy. (left) shows that DeepV2D quickly converges within a small number of iterations. In (right) we see that accuracy consistently improves as more views are added. DeepV2D can be applied to variable numbers of views for a variable number of iterations without retraining.

| ScanNet | Abs Rel ↓ | Sq Rel ↓ | RMSE ↓ | RMSE log ↓ | sc inv ↓ | rot.(deg) ↓ | tr. (deg) ↓ | tr. (cm) ↓ |
|---|---|---|---|---|---|---|---|---|
| DeMoN | 0.231 | 0.520 | 0.761 | 0.289 | 0.284 | 3.791 | 31.626 | 15.50 |
| BA-Net (orig.) | 0.161 | 0.092 | 0.346 | 0.214 | 0.184 | 1.018 | 20.577 | 3.390 |
| BA-Net (5-view) | 0.091 | 0.058 | 0.223 | 0.147 | 0.137 | 1.009 | 14.626 | 2.365 |
| DSO (Engel et al., 2018) | | | | | | 0.925 | 19.728 | 2.174 |
| DSO (fcrn-init) | | | | | | 0.946 | 19.238 | 2.165 |
| Ours (nyu) | 0.080 | 0.018 | 0.223 | 0.109 | 0.105 | 0.714 | 12.205 | 1.514 |
| Ours (scannet) | **0.057** | **0.010** | **0.168** | **0.080** | **0.077** | **0.628** | **10.800** | **1.373** |

Table 2: ScanNet experiments evaluating depth and pose accuracy and cross-dataset generalization. Our approach trained on NYU (ours nyu) outperforms BA-Net despite BA-Net being trained on ScanNet data; training on ScanNet (ours scannet) gives even better performance.

all metrics. The design of our approach is motivated by generalizability. Our network only needs to learn feature matching and correspondence; this experiment indicates that by learning these low level tasks, we can generalize well to new data.

Pose accuracy from DSO Engel et al. (2018) is also included in Table 2. We test DSO using both the default initialization and single-image depth initialization using the output of FCRN (Laina et al., 2016). DSO fails to initialize or loses tracking on some of the test sequences so we only evaluate on sequences where DSO is successful. DSO fails on 335 of the 2000 test sequences while DSO (fcrn-init) fails on only 271.

**SUN3D:** SUN3D (Xiao et al., 2013) is another indoor scenes dataset which we use for comparison with DeepTAM. DeepTAM only evaluates their depth module in isolation using the poses provided by dataset, while our approach is designed to estimate poses during inference. We provide results from our SUN3D experiments in Table 3.

| SUN3D | Training Data | L1-Inv ↓ | L1-Rel ↓ | Sc-Inv ↓ |
|---|---|---|---|---|
| SGM | - | 0.197 | 0.412 | 0.340 |
| DTAM | - | 0.210 | 0.423 | 0.374 |
| DeMoN | S11+RGBD+MVS+**SUN3D** | - | - | 0.146 |
| DeepTAM | MVS+SUNCG+**SUN3D** | 0.054 | 0.101 | 0.128 |
| Ours | NYU | 0.056 | 0.106 | 0.134 |
| Ours | NYU + ScanNet | **0.041** | **0.077** | **0.104** |

Table 3: Results on SUN3D dataset and comparison to DeepTAM. DeepTAM only evaluates depth in isolation and uses the poses from the dataset during inference, while our approach jointly estimates camera poses during inference. We outperform DeepTAM and DeMoN on SUN3D even when we do not use SUN3D data for training.

We cannot train using the same data as DeepTAM since DeepTAM is trained using a combination of SUN3D, SUNCG, and MVS, and, at this time, neither MVS nor SUNCG are publicly available. Instead we train on alternate data and test on SUN3D. We test two different versions of our model; one where we train only on NYU, and another where we train on a combination of NYU and ScanNet data. Our NYU model performs similiar to DeepTAM; When we combine with ScanNet data, we outperform DeepTAM even though DeepTAM is trained on SUN3D and is evaluated with ground truth pose as input.

**KITTI:** The KITTI dataset (Geiger et al., 2013) is captured from a moving vehicle and has been widely used to evaluate depth estimation and odometry. We follow the Eigen train/test split (Eigen et al., 2014), and report results in Table 4. We evaluate using the official ground truth depth maps. We compare to the state-of-the-art single-view methods and also multiview approaches such as BA-Net (Tang & Tan, 2018), and outperform previous methods on the KITTI dataset across all metrics.

| KITTI | Multi | $\delta < 1.25 \uparrow$ | $\delta < 1.25^2 \uparrow$ | $\delta < 1.25^3 \uparrow$ | Abs Rel $\downarrow$ | Sq Rel $\downarrow$ | Sq Rel † $\downarrow$ | RMSE $\downarrow$ | RMSE log $\downarrow$ |
|---|---|---|---|---|---|---|---|---|---|
| DORN | N | 0.945 | 0.988 | 0.996 | 0.069 | 0.300 | - | 2.857 | 0.112 |
| DfUSMC | Y | 0.617 | 0.796 | 0.874 | 0.346 | 5.984 | - | 8.879 | 0.454 |
| BA-Net | Y | - | - | - | 0.083 | - | 0.025 | 3.640 | 0.134 |
| Ours | Y | **0.977** | **0.993** | **0.997** | **0.037** | **0.174** | **0.013** | **2.005** | **0.074** |

Table 4: Results on the KITTI dataset. We compare to state-of-the-art single-image depth network DORN (Fu et al., 2018) and multiview BA-Net (Tang & Tan, 2018). BA-Net reports results using a different form of the Sq-Rel metric which we denote by †.

Overall, the depth experiments demonstrates that imposing geometric constraints on the model architecture leads to higher accuracy and better cross-dataset generalization. By providing a differentiable mapping from optical flow to camera motion, the motion network only needs to learn to estimate interframe correspondence. Likewise, the 3D cost volume means the the depth network only needs to learn to perform stereo matching. These tasks are easy for the network to learn, which leads to strong results on all datasets, and can generalize to new datasets.

## 4.2 TRACKING EXPERIMENTS

DeepV2D can be turned into a basic SLAM system. Using NYU and ScanNet for training, we test tracking performance on the TUM-RGBD tracking benchmark (Table 5) using sensor depth as input. We achieve a lower translational rmse [m/s] than DeepTAM on most of the sequences. DeepTAM uses optical flow supervision to improve performance, but since our network directly maps optical flow to camera motion, we do not need supervision on optical flow.

We use our *global* pose optimization in our tracking experiments. We maintain a fixed window of 8 frames during tracking. At each timestep, the pose of the first 3 frames in the window are fixed and the remaining 5 are updated using the motion module. After the update, the start of the tracking window is incremented by 1 frame. We believe our ability to jointly update the pose of multiple frames is a key reason for our strong performance on the RGB-D benchmark.

| | 360 | desk | desk2 | plant | room | rpy | xyz | mean |
|---|---|---|---|---|---|---|---|---|
| DVO (Kerl et al., 2013) | 0.125 | 0.037 | 0.020 | 0.062 | 0.042 | 0.082 | 0.051 | 0.060 |
| DeepTAM (Zhou et al., 2018) | 0.054 | **0.027** | **0.017** | 0.057 | 0.039 | 0.065 | 0.019 | 0.040 |
| DeepTAM (w/o flow) (Zhou et al., 2018) | 0.069 | 0.042 | 0.025 | 0.063 | 0.051 | 0.070 | 0.030 | 0.050 |
| Ours | **0.046** | 0.034 | **0.017** | **0.052** | **0.032** | **0.037** | **0.014** | **0.033** |

Table 5: Tracking results in the RGB-D benchmark (translational rmse [m/s]).

## 5 CONCLUSION

We propose DeepV2D, a deep learning architecture which is built by composing classical geometric algorithms into a fully differentiable pipeline. DeepV2D is flexible and performs well across a variety of tasks and datasets.

**Acknowledgements** We would like to thank Zhaoheng Zheng for helping with baseline experiments. This work was partially funded by the Toyota Research Institute, the King Abdullah University of Science and Technology (KAUST) Office of Sponsored Research (OSR) under Award No. OSR-2015-CRG4-2639, and the National Science Foundation under Grant No. 1617767.

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

## A  APPENDIX

### A.1  LS-OPTIMIZATION LAYER:

In Equation 4 we defined the residual error to be:

$$\mathbf{e}_k^{ij}(\xi_i, \xi_j) = \mathbf{r}_k - [\pi((e^{\xi_j}\mathbf{G}_j)(e^{\xi_i}\mathbf{G}_i)^{-1}\mathbf{X}_k^i) - \pi(\mathbf{G}_{ij}\mathbf{X}_k^i)], \qquad \mathbf{X}_k^i = \pi^{-1}(\mathbf{x}_k, z_k) \qquad (8)$$

and the objective function as the weighted sum of error terms:

$$E(\boldsymbol{\xi}) = \sum_{(i,j)\in\mathcal{C}} \sum_k \mathbf{e}_k^{ij}(\xi_i, \xi_j)^T \, diag(\mathbf{w}_k) \, \mathbf{e}_k^{ij}(\xi_i, \xi_j), \qquad diag(\mathbf{w}_k) = \begin{pmatrix} w_k^u & 0 \\ 0 & w_k^v \end{pmatrix} \qquad (9)$$

We apply a Gauss-Newton update to Equation 9. The Gauss-Newton update is computed by solving for the minimum of the second order approximation of the objective function:

$$\xi^* = -(\mathbf{J}^T\mathbf{W}\mathbf{J})^{-1}\mathbf{J}^T\mathbf{W}\mathbf{r}(\xi_1, ..., \xi_N), \qquad \mathbf{J}_p = \frac{\partial r_p(\boldsymbol{\epsilon})}{\partial \boldsymbol{\epsilon}}\Big|_{\boldsymbol{\epsilon}=\mathbf{0}} \qquad (10)$$

where $\mathbf{r}(\xi_1, ..., \xi_N)$ is the stack of residuals and $\mathbf{J}$ is the Jacobian matrix. Each row $\mathbf{J}_i$ is the Jacobian of the $i^{th}$ error term w.r.t to each of the parameters. Each $\xi$ is 6-dimensional, so optimizing over $N$ poses means we are updating $6N$ variables.

Let $r_p = e_k^{ij}(\xi_i, \xi_j)$ be the p$^{th}$ residual, then

$$\frac{\partial e_k^{ij}(\xi_i, \xi_j)}{\partial \xi_j}\Big|_{\xi_i=0, \xi_j=0} = \frac{\partial}{\partial \xi_j}[\mathbf{r}_k - [\pi((e^{\xi_j}\mathbf{G}_j)(e^{\xi_i}\mathbf{G}_i)^{-1}\mathbf{X}_k^i) - \pi(\mathbf{G}_{ij}\mathbf{X}_k^i)]] =$$

$$\frac{\partial}{\partial \xi_j}\pi((e^{\xi_j}\mathbf{G}_j)(e^{\xi_i}\mathbf{G}_i)^{-1}\mathbf{X}_k^i) = \frac{\partial}{\partial \xi_j}\pi(e^{\xi_j}(\mathbf{G}_{ij}\mathbf{X}_k^i)) \qquad (11)$$

$$\frac{\partial e_k^{ij}(\xi_i, \xi_j)}{\partial \xi_j}\Big|_{\xi_i=0, \xi_j=0} = \frac{\partial}{\partial(\mathbf{G}_{ij}\mathbf{X}_k^i)}\pi(\mathbf{G}_{ij}\mathbf{X}_k^i) \cdot \frac{\partial}{\partial \xi_j}e^{\xi_i}(\mathbf{G}_{ij}\mathbf{X}_k^i)$$

Likewise, the Jacobian for $\xi_j$ is

$$\frac{\partial e_k^{ij}(\xi_i, \xi_j)}{\partial \xi_j}\Big|_{\xi_i=0, \xi_j=0} = \frac{\partial}{\partial \xi_i}[\mathbf{r}_k - [\pi((e^{\xi_j}\mathbf{G}_j)(e^{\xi_i}\mathbf{G}_i)^{-1}\mathbf{X}_k^i) - \pi(\mathbf{G}_{ij}\mathbf{X}_k^i)]] =$$

$$\frac{\partial}{\partial \xi_i}\pi((e^{\xi_j}\mathbf{G}_j)(e^{\xi_i}\mathbf{G}_i)^{-1}\mathbf{X}_k^i) = \frac{\partial}{\partial \xi_i}\pi(\mathbf{G}_j\mathbf{G}_i^{-1}e^{-\xi_i}\mathbf{X}_k^i) = \frac{\partial}{\partial \xi_i}\pi(\mathbf{G}_{ij}e^{-\xi_i}\mathbf{X}_k^i) \qquad (12)$$

using the adjoint to move the increment to the left of the transformation

$$= \frac{\partial}{\partial \xi_i} \pi(e^w \mathbf{G}_{ij} \mathbf{X}_k^i) \qquad \text{where } w = -Adj_{\mathbf{G}_{ij}} \cdot \xi$$

$$\frac{\partial e_k^{ij}(\xi_i, \xi_j)}{\partial \xi_j}|_{\xi_i=0, \xi_j=0} = -\frac{\partial}{\partial(\mathbf{G}_{ij}\mathbf{X}_k^i)} \pi(\mathbf{G}_{ij}\mathbf{X}_k^i) \cdot \frac{\partial}{\partial w} e^w(\mathbf{G}_{ij}\mathbf{X}_k^i) \cdot Adj_{\mathbf{G}_{ij}}$$

(13)

where the Jacobian of the action of a $\mathbf{SE}(3)$ element on a 3D point is computed

$$\frac{\partial e^\xi \mathbf{X}}{\partial \xi}|_{\xi=0} = \left[ \begin{array}{ccc|ccc} 1 & 0 & 0 & 0 & -Z & Y \\ 0 & 1 & 0 & Z & 0 & X \\ 0 & 0 & 1 & -Y & X & 0 \end{array} \right]$$

(14)

During training, we propagate through the Gauss-Newton update. The update is found by solving the linear system

$$\mathbf{H}\xi = -\mathbf{b}, \qquad \mathbf{H} = \mathbf{J}^T \mathbf{W} \mathbf{J}, \; \mathbf{b} = \mathbf{J}^T \mathbf{W} \mathbf{r}(\xi_1, ..., \xi_N)$$

(15)

Since $\mathbf{H}$ is positive definite, we solve Equation 15 using Cholesky decomposition. In the backward pass, the gradients can be found by solving another linear system.

$$\frac{\partial \mathcal{L}}{\partial \mathbf{H}} = -(\mathbf{H}^{-1}\frac{\partial \mathcal{L}}{\partial \xi})^T \xi, \qquad \frac{\partial \mathcal{L}}{\partial \mathbf{b}} = \mathbf{H}^{-1}\frac{\partial \mathcal{L}}{\partial \xi}^T$$

(16)

## B  TRAINING DETAILS

DeepV2D is implemented in Tensorflow (Abadi et al., 2016). All components of the network are trained from scratch without using any pretrained weights. We use gradient checkpointing (Chen et al., 2016) to reduce memory usage and increase batch size.

When training on NYU and ScanNet, we train with 4 frame video clips. On KITTI, we use 5 frame video clips. The video clips are created by first selecting a keyframe. The other frames are randomly sampled from the set of frames within a specified time window of the keyframe. For example, on NYU, we create the training video by sampling from frames within 1 second of the keyframe.

Training occurs in the following two stages:

**Stage I:** We train the Motion Module using the $L_{motion}$ loss with RMSProp (Tieleman & Hinton, 2012) and a learning rate of 0.0001. For the input depth, we use the ground truth depth with missing values interpolated. We train Stage I for 20k iterations on NYU, 16k iterations on KITTI, and 30k iterations on ScanNet.

**Stage II:** In stage II, we jointly train the motion and depth modules end-to-end on the combined loss with RMSProp. The initial learning rate is set to .001 and decayed to .0002 after 100k training steps. During the second stage we store depth predictions to be used during the next training epoch. We train Stage II for a total of 120k iterations with a batch size of 2. In our ScanNet experiments, we train for an additional 60k iterations.

**Data Augmentation:**  We perform data augmentation by adjusting brightness, gamma, and performing random scaling of the image channels. We also randomly perturb the input camera pose to the Motion Module by sampling small perturbations.

## C  TIMING AND MEMORY USAGE

In the below table we provide timing and peak memory usage for different versions of our method. All results are obtained using 8 frame video sequences as input with the exception of the basline single-image network FCRN Laina et al. (2016) which uses a single frame as input.

|  | Abs-Rel ↓ | Parameters | Peak GPU Memory | Iteration Time |
|---|---|---|---|---|
| FCRN (Laina et al., 2016) | 0.121 | 64M | 0.1G | 0.05s |
| Ours (1/2 res) | 0.083 | 32M | 0.7G | 0.22s |
| Ours (1-HG) | 0.071 | 16M | 2.8G | 0.61s |
| Ours (corr) | 0.135 | 25M | 1.8G | 0.32s |
| Ours | 0.062 | 32M | 2.8G | 0.69s |

Table 6: Timing and memory details for different versions of our approach.

In ours(1-HG) we replace the feature extractor with a single 2D-hourglass network, and replace the stereo network with a single 3D-hourglass network. The shallower network still performs well, but causes Abs-Rel to increase from 0.065 to 0.071, showing that stacking hourglass networks is beneficial for performance. In ours (1/2 res) we test the performance of DeepV2D when images are downsampled to 1/2 resolution for training and inference. Using lower resolution images decreases memory usage and inference time but slightly decreases accuracy.

We also test a version where we replace the 3d stereo network with a correlation layer and 2d encoder-decoder. In ours(corr), we take the correlation between features over the same depth range as we use to build the 3D cost volume, then concatenate the correlation response with features from the keyframe image, similar to DispNet (Mayer et al., 2016b). The correlation version performs worse, increasing Abs-Rel from 0.065 to 0.135. This is consistent with prior work which has demonstrated that 3D cost volumes give better performance than direct correlation (Kendall et al., 2017; Chang & Chen, 2018).

## D   ADDITIONAL TRACKING INFORMATION

In Table 7 we report tracking results for all sequences in the Freiburg 1 dataset.

| Sequence | RGB-D SLAM | DeepTAM | Ours |
|---|---|---|---|
| 360 | 0.119 | 0.063 | **0.056** |
| 360(v) | 0.125 | 0.054 | **0.046** |
| desk | 0.030 | 0.033 | **0.029** |
| desk(v) | 0.037 | **0.027** | 0.034 |
| desk2 | 0.055 | 0.046 | **0.041** |
| desk2(v) | 0.020 | **0.017** | **0.017** |
| floor | 0.090 | 0.081 | **0.064** |
| plant | 0.036 | 0.027 | **0.019** |
| plant(v) | 0.062 | 0.057 | **0.052** |
| room | 0.048 | **0.040** | 0.047 |
| room(v) | 0.042 | 0.039 | **0.032** |
| rpy | 0.043 | 0.046 | **0.039** |
| rpy(v) | 0.082 | 0.065 | **0.037** |
| teddy | 0.067 | 0.059 | **0.043** |
| xyz | 0.051 | **0.019** | 0.025 |
| xyz(v) | 0.024 | 0.017 | **0.016** |
| Average | 0.058 | 0.043 | **0.037** |

Table 7: Per-Sequence tracking results on the RGB-D benchmark evaluated using translational RMSE [m/s]. We outperform DeepTAM and DVO on 12 of the 16 sequences and achieve a lower translational RMSE averaged over all sequences. While DeepTAM requires optical flow supervision to achieve good performance, we do not require supervision on optical flow since the relation between camera motion and optical flow is embedded into our network architecture.

# E  CAMERA POSE ABLATIONS

The focus of this work on depth estimation, but we are interested in how different methods for estimating camera pose impact the final performance. In Table 8, we test different methods for estimating camera pose on NYU. In each experiment, we replace the motion module of our trained network with the given alternative, and test the final results. We also report results from MVSNet (trained on NYU) using each SfM implementation.

COLMAP (Schonberger & Frahm, 2016a) and OpenMVG (Moulon et al.) are publicly available SfM implementations. They do not return results on all input sequences, so we only evaluate sequences were they converge without an error. PWCNet+Ceres takes the output of an optical flow network, PWCNet (Sun et al., 2018), and performs joint optimization of depth and pose using the Ceres solver (Agarwal et al., 2012). Finally, we evaluate MVSNet (Yao et al., 2018) when the pose predicted by DeepV2D is given as input. Note that not all SfM implementations converge on all sequences (success rate is reported in parenthesis) and we only evaluate the method on the frames in which it converges.

| Depth | Motion | Abs-Rel $\downarrow$ | $\delta_1 \uparrow$ | $\delta_2 \uparrow$ | $\delta_3 \uparrow$ |
|---|---|---|---|---|---|
| MVSNet | Identity | 0.419 | 0.382 | 0.681 | 0.859 |
| DeepV2D | Identity | 0.362 | 0.460 | 0.756 | 0.901 |
| MVSNet | COLMAP (274/654) | 0.244 | 0.724 | 0.857 | 0.925 |
| DeepV2D | COLMAP | 0.199 | 0.741 | 0.878 | 0.940 |
| MVSNet | OpenMVG (422/654) | 0.181 | 0.766 | 0.913 | 0.965 |
| DeepV2D | OpenMVG | 0.173 | 0.774 | 0.913 | 0.963 |
| MVSNet | PWC+Ceres (654/654) | 0.279 | 0.651 | 0.845 | 0.925 |
| DeepV2D | PWC+Ceres | 0.274 | 0.664 | 0.846 | 0.925 |
| MVSNet | DeepV2D (654/654) | 0.101 | 0.885 | 0.970 | 0.990 |
| | DeepV2D (ours) | **0.062** | **0.955** | **0.990** | **0.996** |

Table 8: Impact of pose estimation method on depth accuracy. Replacing our motion module with SfM degrades performance for both MVSNet and our approach.

We also show results of our method when the motion module is replaced with other methods for estimation motion. In all cases, using SfM results in worse performance. We observe that classical SfM is not robust enough to consistently produce accurate poses, which leads to large errors on the test set. MVSNet performs better using the poses estimated by our network, but still underperforms our full system, showing the importance of differentiable alternation between pose and stereo.

## F    ADDITIONAL RESULTS

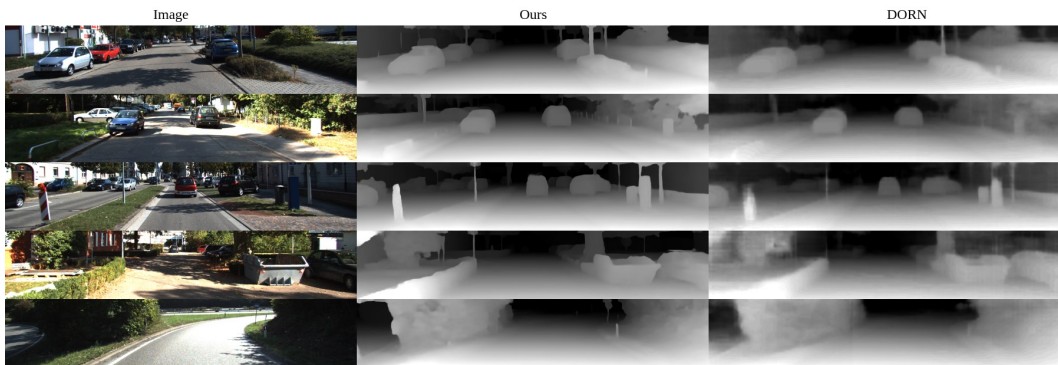

Figure 6: Visualizations of depth predictions on KITTI dataset.

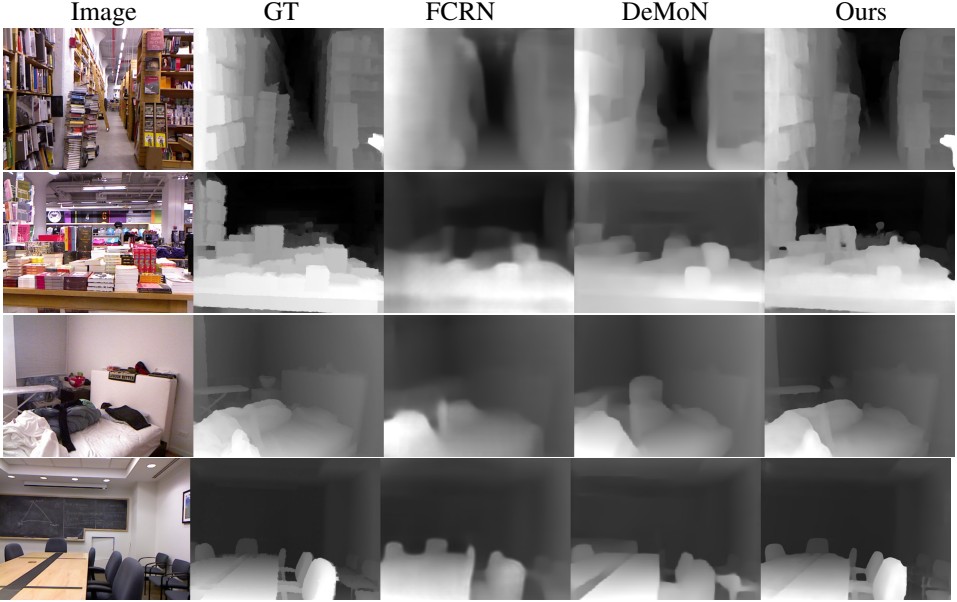

Figure 7: Additional results on the NYU depth dataset Silberman et al. (2012) using 7-frame video clips. We show results compared with Laina et al. (2016) and Ummenhofer et al. (2017).

## G NETWORK ARCHITECTURES

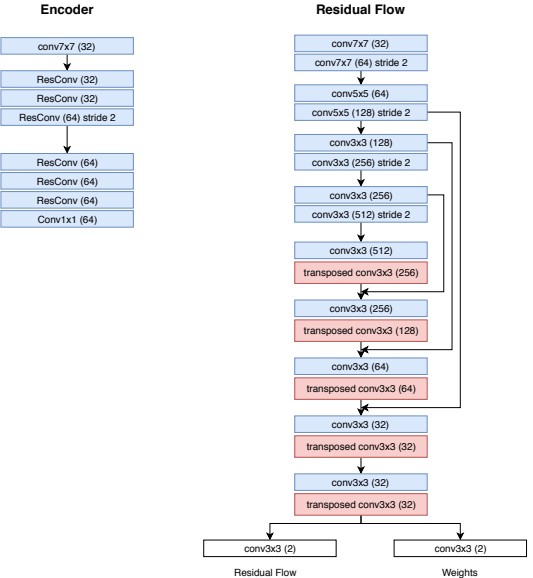

Figure 8: Motion Module Architecture: The Encoder(left) extracts a dense 1/4 resolution feature map for each of the input images. The Residual Flow Network (right) takes in a pair of feature maps and estimates the residual flow and corresponding weights. This residual flow is estimated with an encoder-decoder network, with skip connections formed by concatenating feature maps. Numbers in parenthesis correspond to the number of output channels for each layer.

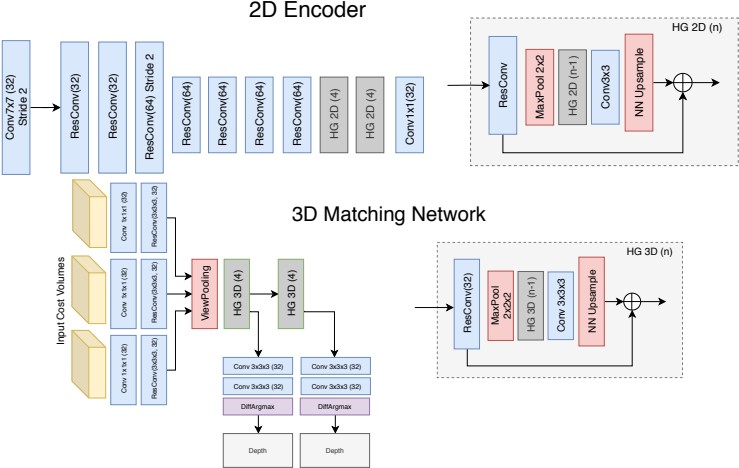

Figure 9: Depth Module Architecture: The 2D encoder (top) is applied to each image in the video sequence. The 2D Encoder consists of a series of residual convolutions and 2 Hourglass Networks. The hourglass networks process the incoming features maps as multiple scales. The hourglass network is defined recursively (i.e. HG(n) contains lower resolution hourglass HG(n-1)). We use 4 nested hourglass modules with feature dimension 64-128-192-256. The resulting feature maps from the 2D encoder are used to construct the cost volumes. The 3D matching network (bottom) takes a collection of cost volumes as input. After a 1x1x1 convolutional layer and a 3x3x3 residual convolution, we perform view pooling, which aggregates information over all the frames in the video. The aggregated volume is then processed by a series of 3D hourglass networks, each of which outputs an intermediate depth estimate. The widths of the 3D hourglass is 32-80-128-176.

