# OpenReview forum: "DeepV2D: Video to Depth with Differentiable Structure from Motion"
_ICLR.cc/2020/Conference — Accept (Poster)_

### Official Review · AnonReviewer2 · 2019-10-22
**Official Blind Review #2**

**Rating:** 8

**Review:**

This work proposes a neural network architecture for joint depth and camera motion estimation on video sequences. The authors propose an architecture that incorporates classic principles from SfM, namely depth computation based on cost volumes and motion estimation based on the reprojection error of features. Extensive experiments on a variety of datasets are shown and support that this approach provides strong results.

+ Principled approach that marries the best aspects of deep learning with classic principles from multi-view geometry.

+ Well-written paper

+ Generalizes well

+ Significantly outperforms the state of the art

+ Clearly shows that more views help

+ Seems to be robust to initialization


Question: What happens if more Gauss-Newton steps are made? Could one trade computation for quality here?


Minor issues:

- Equation (1), right: f_z should probably be f_x
- Paragraph between Eq. (1) and (2): There seems to be something wrong with the typesetting of x^i=... (e.g. the equals sign)
- Same for the paragraph before Eq. (1) and x=(u,v)


Summary: This paper presents a well-engineered and non-trivial system that leverages principles from different fields in a very reasonable way. The results look great both qualitatively and quantitatively. The experiments are extensive and show a clear improvement over the state-of-the-art.


**Experience Assessment:**

I have published one or two papers in this area.

**Review Assessment: Checking Correctness Of Derivations And Theory:**

I assessed the sensibility of the derivations and theory.

**Review Assessment: Checking Correctness Of Experiments:**

I assessed the sensibility of the experiments.

**Review Assessment: Thoroughness In Paper Reading:**

I read the paper at least twice and used my best judgement in assessing the paper.

---

> ### Author Response · Authors · 2019-11-15
> **Thank you for your review and comments.**
>
> The minor issues are corrected in the revision, thank you for pointing us to these typos.
>
> Q. What happens if more Gauss-Newton steps are made?
>
> Using more Gauss-Newton steps actually doesn’t give any increase in performance (increasing from 1 to 3 gives the same MRE on NYU). The reason is that minimizing geometric reprojection error is a relatively easy optimization problem. When camera motion is small, the second order approximation of the objective function is very good, so a single step is sufficient.
>
> However, as shown in Figure 5 (left), we can improve efficiency with a slight loss of accuracy by using fewer global iterations (where each iteration is a full pass of the motion and depth networks).

---

### Official Review · AnonReviewer1 · 2019-10-28
**Official Blind Review #1**

**Rating:** 6

**Review:**

The authors proposed to estimate depth from a video sequence. In the model, the pipeline iteratively estimates motion and depth by separate modules, which can be trained in an end-to-end fashion. The numerical results show better scores than the other SOTA methods.  Overall, I think this is a useful work and may be considered for publishing.

The following are the detailed comments:
1. The introduction of the related works is well written.
2. The empirical comparison is quite thorough and demonstrates the proposed method using sequence and multi-frame is very useful.
3. The method has good generalization.

4. Besides the empirical results, I found the paper reads quite mechanical and provides very limited understanding if any. It would be much better if the authors can make further effort to understand where exactly the performance comes from. With a more complete story, the paper would have a better potential to last longer.

**Experience Assessment:**

I have read many papers in this area.

**Review Assessment: Checking Correctness Of Derivations And Theory:**

I assessed the sensibility of the derivations and theory.

**Review Assessment: Checking Correctness Of Experiments:**

I assessed the sensibility of the experiments.

**Review Assessment: Thoroughness In Paper Reading:**

I read the paper at least twice and used my best judgement in assessing the paper.

---

> ### Author Response · Authors · 2019-11-15
> **Thank you for your review**
>
>  In the revision, we’ve added more analysis of the experiments to provide more understanding of where the performance gain is coming from. The performance of our method is from a combination of factors, including improvements in both depth and motion, and how they are combined into a single network. The improvement in motion comes from embedding a least-squares optimization layer in the network which jointly updates the poses of all cameras, which gives better results than using generic network layers to predict camera motion (Table 5).  We achieve good depth by adapting a stereo network (Kendall et al., 2017) for monocular video, and show that alternating motion and depth updates at inference time converges to accurate depth and motion (Tables 1-4).

---

### Official Review · AnonReviewer3 · 2019-11-01
**Official Blind Review #3**

**Rating:** 6

**Review:**

This paper proposes a framework for training machine learning models that simultaneously estimates depth of objects and poses of a single camera in a sequence of images from a single camera, in others words, a video.
In a video, to estimate depth of objects, which is a main objective of this paper, we need to know the relative position and rotation information of a single camera in a sequence of images in a video (motional information). However, to estimate the relative position and rotation information of the camera, we need to know a ground truth depth information of each objects in each image.
The main idea works just like EM or alternating optimization. The depth module estimates depth of objects in a sequence of images assuming the relative position and rotation information of a single camera is given. The motion module estimates motional information of a camera assuming the depth of each object is given.
The authors formulate the aforementioned two modules as neural networks, so that they can be trained end-to-end, and proposes various way of initializing the two modules.

While the idea is simple, I think the paper is well-written and the experiments show a superior performance over existing approaches.


**Experience Assessment:**

I do not know much about this area.

**Review Assessment: Checking Correctness Of Derivations And Theory:**

I assessed the sensibility of the derivations and theory.

**Review Assessment: Checking Correctness Of Experiments:**

I assessed the sensibility of the experiments.

**Review Assessment: Thoroughness In Paper Reading:**

I read the paper at least twice and used my best judgement in assessing the paper.

---

> ### Author Response · Authors · 2019-11-15
> **Thank you for your review**
>
> Thank you for your review and comments.

---

### Official Review · AnonReviewer4 · 2019-11-04
**Official Blind Review #4**

**Rating:** 6

**Review:**

This paper pushes forward the research of deep learning based video 3D reconstruction by decomposing the problem in two-stages:
1. Depth estimation from multi-view stereo
2. Camera pose estimation from optical flow estimation and PnP SE3 pose,
which turns out to achieve state-of-the-art performance on public datasets.

In general, the iterative procedure of this paper is similar to DeepTAM (Zhou et al., 2018), the major difference is that the camera pose is estimated by PnP with estimated Flow but not directly predicted from sub-network structures, which contributes to the higher tracking accuracy and further improves the accuracy of multi-view stereo depth estimation.

A multi-view camera pose estimation (the global pose optimization in the paper) is also proposed to utilize the relative relation between all video frame pairs, which is difficult for fully CNN based pipeline and maximizes the use of the explicit PnP optimization.

After that, a residual flow field is predicted using CNNs and camera poses is further refined by minimizing the geometric re-projection error.

So this paper focuses more on the camera pose estimation than the depth, which is a good start point to achieve better multi-view capabilities in a CNN framework.


However, I still have several concerns for this paper:

1. The full system is highly engineered and complicated. For example, the feature map is extracted from two hour-glass networks, which seems over-complicated for feature extraction, and the multi-view stereo network using four 3D hour-glass networks, which consumes a large amount of memories. So I would like to see the authors demonstrate the performance from in simpler settings, e.g. the feature map can be a single encoder-decoder and the multi-view stereo is done by correlation and 2D convolution. So I would like to see the inference time, peak memory consumption and the model size. Ablation studies will also help the readers to understand whether performance gain is from the pose estimation or the network capacities, which is unclear in the current paper (Appendix.D).

2. The comparison with state-of-the-art conventional system is missing. For example, is the camera pose estimation better than initializing the DSO (Engel et al., 2018) with a monocular depth estimation? In real applications, if the performance gain is insignificant, the conventional method will still be a better choice because the CNN based methods are computationally expensive on platforms without powerful GPUs. I will not downgrade the rating if the performance gain is insignificant, but it is necessary to see the comparison.


3. Even the difference is ignorable for performance, I hope the author could use adjoint when deriving the derivatives in Eq.(11) of Appendix A.1. The author can refer to this tutorial http://ethaneade.com/lie.pdf or the text book https://www.eecis.udel.edu/~cer/arv/readings/old_mkss.pdf.

PS: For the current version with 10 pages, I lean to a borderline score, but I select 6 because 5 is not an option. I will keep or raise the current score if my concerns are addressed during rebuttal.

**Experience Assessment:**

I have published in this field for several years.

**Review Assessment: Checking Correctness Of Derivations And Theory:**

I carefully checked the derivations and theory.

**Review Assessment: Checking Correctness Of Experiments:**

I carefully checked the experiments.

**Review Assessment: Thoroughness In Paper Reading:**

I read the paper at least twice and used my best judgement in assessing the paper.

---

> ### Author Response · Authors · 2019-11-15
> **Thank you for your review and suggestions.**
>
> Thank you for your review and suggestions. We have submitted a revised version following your suggestions. Below we address individual points.
>
> 1. Our overall approach is quite modular and components can easily be swapped depending on the application. We add Appendix C in our revision which demonstrates several versions along with parameter counts, timing information, peak memory usage, and depth accuracy. In terms of raw parameter counts, our method uses fewer parameters than the single-image depth baseline (Laina et al., 2016).
> Following your suggestions, we implement a simpler version of our model (1-HG) where we replace the feature extractor with a single 2d-hourglass network and replace the stereo network with a single 3d-hourglass network. The results in Table 6 show that this causes Abs-Rel to increase from 0.065 to 0.071, which is still significantly better than prior work.
>
> We also test a version where we replace the 3d stereo network with a correlation layer and 2d encoder-decoder (Ours (corr) Table 6). We take the correlation between features over the same depth range as we use to build the 3D cost volume, then concatenate the correlation response with features from the keyframe image, similar to DispNet (Mayer et al., 2016). The correlation version performs worse,  increasing Abs-Rel from 0.065 to 0.135. This is consistent with prior work which has demonstrated that 3D cost volumes give better performance than direct correlation (Kendall et al., 2017; Chang & Chen, 2018).
>
> 2. “Comparison to DSO”.  We’ve added pose results from DSO to Table 2. We test DSO on ScanNet because the TUM-RGBD dataset is captured using a rolling shutter camera . DSO fails to initialize or diverges on 335/2000 of the videos, so we only report results on the test sequences where tracking is successful. Using single-image depth for initialization improves the performance of DSO, reducing the failure rate from 335 to 271 of the 2000 videos and giving slightly better pose metrics, but the estimated poses are still less accurate than our method; the rotation and translational angular error of DSO is (R: 0.946, T: 19.238) while our method gets significantly more accurate poses (R: 0.628, T: 10.800).
>
> 3. Thank you for this suggestion, we’ve updated the derivation in Eq. 12 to use the adjoint.

---

### Decision · Program_Chairs · 2019-12-19

**Decision:**

Accept (Poster)

**Comment:**

This work proposes a CNN architecture for joint depth and camera motion estimation from videos. The paper presents a differentiable formulation of the problem to allow its end-to-end learning, and the reviewers unanimously find the proposed approach reasonable and agree that this is a solid paper. Some of the reviewers find the method itself to be too mechanical, but they all agree that this is a well-engineered solution.